# Prognostic Value of the Diversity of Nuclear Chromatin Compartments in Gynaecological Carcinomas

**DOI:** 10.3390/cancers12123838

**Published:** 2020-12-19

**Authors:** Andreas Kleppe, Fritz Albregtsen, Jone Trovik, Gunnar B. Kristensen, Håvard E. Danielsen

**Affiliations:** 1Institute for Cancer Genetics and Informatics, Oslo University Hospital, NO-0424 Oslo, Norway; andrekle@ifi.uio.no (A.K.); fritz@ifi.uio.no (F.A.); gbk@ous-hf.no (G.B.K.); 2Department of Informatics, University of Oslo, NO-0316 Oslo, Norway; 3Department of Obstetrics and Gynecology, Haukeland University Hospital, NO-5020 Bergen, Norway; jone.trovik@helse-bergen.no; 4Department of Clinical Science, University of Bergen, NO-5020 Bergen, Norway; 5Department of Gynecologic Oncology, Oslo University Hospital, NO-0424 Oslo, Norway; 6Institute of Clinical Medicine, University of Oslo, NO-0318 Oslo, Norway; 7Nuffield Division of Clinical Laboratory Sciences, University of Oxford, Oxford OX3 9DU, UK

**Keywords:** chromatin compartments, chromatin organisation, texture analysis, prognostic marker, ovarian cancer, endometrial cancer

## Abstract

**Simple Summary:**

Chromatin organisation affects gene expression and contributes to carcinogenesis. Automatic quantification of chromatin heterogeneity can be applied to identify patients with increased risk of cancer recurrence and death in several cancer types. We aimed to investigate the prognostic role of diversity of chromatin compartments in relation to chromatin heterogeneity and as a potential supplement to pathological risk classifications in gynaecological carcinomas. To this end, we computed the entropy of both chromatin compartment sizes and optical densities within compartments. In analysis of two cohorts consisting of 1037 patients with gynaecological carcinoma, we observed a moderately strong correlation between the prognostic value of the entropies and chromatin heterogeneity. The entropies provided an objective marker, which, integrated with pathological risk classifications, might possibly contribute to the selection of high-risk stage I ovarian carcinoma patients for adjuvant chemotherapy and to preoperative identification of low-risk endometrial carcinoma patients who are candidates for less extensive surgery.

**Abstract:**

Statistical texture analysis of cancer cell nuclei stained for DNA has recently been used to develop a pan-cancer prognostic marker of chromatin heterogeneity. In this study, we instead analysed chromatin organisation by automatically quantifying the diversity of chromatin compartments in cancer cell nuclei. The aim was to investigate the prognostic value of such an assessment in relation to chromatin heterogeneity and as a potential supplement to pathological risk classifications in gynaecological carcinomas. The diversity was quantified by calculating the entropy of both chromatin compartment sizes and optical densities within compartments. We analysed a median of 281 nuclei (interquartile range (IQR), 273 to 289) from 246 ovarian carcinoma patients and a median of 997 nuclei (IQR, 502 to 1452) from 791 endometrial carcinoma patients. The prognostic value of the entropies and chromatin heterogeneity was moderately strongly correlated (*r* ranged from 0.68 to 0.73), but the novel marker was observed to provide additional prognostic information. In multivariable analysis with clinical and pathological markers, the hazard ratio associated with the novel marker was 2.1 (95% CI, 1.3 to 3.5) in ovarian carcinoma and 2.4 (95% CI, 1.5 to 3.9) in endometrial carcinoma. Integration with pathological risk classifications gave three risk groups with distinctly different prognoses. This suggests that the novel marker of diversity of chromatin compartments might possibly contribute to the selection of high-risk stage I ovarian carcinoma patients for adjuvant chemotherapy and low-risk endometrial carcinoma patients for less extensive surgery.

## 1. Introduction

The higher-order chromatin organisation in human cell nuclei has been linked to gene expression and found to alter during cell differentiation [1,2]. Defects in the higher-order chromatin organisation can result in human diseases [3]. In human cancer cells, chromatin organisation has been observed to be a dominant determinant of variation in regional mutation rates [4,5].

Statistical texture analysis of cancer cell nuclei stained for DNA has recently been used to develop a marker of chromatin heterogeneity that quantifies the level of disorganisation in the chromatin structure [6]. The marker has been shown to identify patients with increased risk of cancer-specific death in several cancer types [6] and is externally validated by independent investigators to predict overall and disease-free survival in colorectal cancer [7].

The marker of chromatin heterogeneity quantifies the chromatin entropy using the extension of the grey level entropy matrix (GLEM) [8] to four dimensions (GLEM4D) [9]. In the development of the marker, patterns of aberrant chromatin organisation were identified by mapping subregions of nuclear images to elements in the GLEM4D and then correlating the frequency of chromatin patterns residing to a GLEM4D element to the patient outcome observed in the training dataset [6]. This provided a score quantifying whether and to which extent a chromatin pattern indicates good or poor clinical outcome, which was used to classify new patients as chromatin homogeneous (CHO) or chromatin heterogeneous (CHE) [6,7]. As high entropy in subregions of nuclear images was associated with poor clinical outcome, many interleaved chromatin compartments with different condensations will cause the patient to be classified as CHE [6]. This indirect analysis of chromatin compartments will however ignore whether the compartments are more or less similar throughout the nucleus in terms of some attributes, e.g., volume, number, peripheral localisation or shape.

In this study, we aimed to investigate the prognostic value of a more direct analysis of chromatin compartments by automatically quantifying the diversity of chromatin compartments in cancer cell nuclei. As the size and number of chromatin compartments have been observed to vary during carcinogenesis [10,11,12,13,14,15], we computed the entropy of chromatin compartment sizes because an increase in size or number of chromatin compartments is expected to increase this entropy. The entropy was coupled with the entropy of the measured optical densities within the chromatin compartments, which suggests the diversity of DNA content in the chromatin compartments of a cancer cell nucleus. The entropies were computed separately for highly and weakly condensed compartments. The relatively weakly condensed compartments are assumed to be euchromatic regions that are transcriptionally active, while the relatively highly condensed compartments are assumed to be inactive, heterochromatic regions in the nucleus. Adaptive feature values were computed by correlating the entropies with the clinical outcome after surgical removal of gynaecological carcinomas. The feature values were applied to develop an objective marker of diversity of chromatin compartments using a set of patients and then evaluated on another set of patients. The feature values of this novel marker were compared to the feature value of the marker of chromatin heterogeneity, and we also investigated whether the novel marker might improve risk stratification of patients with ovarian or endometrial carcinoma.

## 2. Materials and Methods

### 2.1. Patient Cohorts

The first cohort consisted of 246 patients with International Federation of Gynecology and Obstetrics (FIGO) stage I ovarian carcinoma that were treated during 1982–1989 [16]. All patients had their ovarian cancer resected in a surgical procedure that involved peritoneal washing, hysterectomy, bilateral salpingo-oophorectomy and omentectomy. Surgery was generally performed at county hospitals in Norway and followed by evaluation of further treatment at the Norwegian Radium Hospital. The FIGO stage was reviewed according to 1988 criteria, albeit lymphadenectomy was not routinely performed. All patients provided informed consent, and the study was in accordance with Norwegian law.

A cohort of 791 consenting patients from the multicentre trial Molecular Markers in Treatment of Endometrial Cancer (MoMaTEC) (clinical trial identifier NCT00598845) were prospectively included in this study [17]. These patients were treated for endometrial carcinoma between 2001 and 2011 and staged according to 2009 criteria [18]. The Regional Committees for Medical and Health Research Ethics (REK) in Norway approved the study (REKIII no. 052.01).

Each patient cohort was divided into a train subset and a test subset, and it was verified that the subsets were roughly balanced on surgery date and important clinical parameters. The subsets were the same as in previous studies that analysed the same patient cohorts [9,19]. In the ovarian carcinoma cohort, the train subset consisted of 134 patients and the test subset of 112 patients [9]. In the endometrial carcinoma cohort, the train subset consisted of 398 patients and the test subset of 393 patients [19].

The definition of dichotomous clinical outcome also followed the previous studies [9,19]. For ovarian carcinoma patients, poor outcome was defined as recurrence or cancer-specific death within ten years after surgery. Those who survived ten years without recurring were defined as having good outcome. For endometrial cancer patients, poor outcome was defined as cancer-specific death, and all other patients were defined as having good outcome. There were 72 poor outcome patients in the ovarian carcinoma cohort (37 in the train subset and 35 in the test subset) and 84 in the endometrial carcinoma cohort (44 in the train subset and 40 in the test subset).

### 2.2. Sample Preparation

Tissue samples of the resected ovarian tumours and of curettage specimens from the endometrial carcinoma patients were analysed. First, the samples were fixed in 4% buffered formaldehyde and paraffin-embedded (FFPE). A pathologist selected a representative region for each tumour using haematoxylin and eosin-stained sections. One or more 50 µm sections of each selected tumour region were enzymatically digested (SIGMA protease, type XXIV, Sigma Chemical C., St. Louis, MO, USA) to isolate the cell nuclei. The nuclei were stained by Feulgen-Schiff according to an established protocol [20], and thus exclusively highlighting DNA.

The stained nuclei were imaged by a Zeiss Axioplan light microscope equipped with a Zeiss objective lens (40× magnification, 0.75 numerical aperture), a 546 nm green filter and a black-and-white high-resolution digital camera (C4742-95, Hamamatsu Photonics, Hamamatsu, Japan, was used for the ovarian carcinoma material and AxioCam MrM, Zeiss, Jena, Germany, for the endometrial carcinoma material), providing a depth of field of about 1.5 µm and a physical resolution of about 165 nm per pixel. The images were stored with 10 bits pixel depth and corrected for background effects by utilising an image acquired from an empty region on the slide.

Subimages of whole, isolated, and epithelial cell nuclei were selected by trained personnel for the ovarian carcinoma patients and segmented using a manually chosen global threshold. For the endometrial carcinoma patients, segmentation of each nucleus from the background was completed automatically using the Ploidy Work Station (Room4, Sussex, UK). This software was also used to automatically discard non-intact nuclei (e.g., cut, folded, or connected), necrotic nuclei and non-epithelial nuclei for the endometrial carcinoma patients. Trained personnel verified the automatic nucleus classifications. Any holes in the segmentation mask of a nucleus were removed. The images of the included segmented nuclei are subsequently referred to as nuclear images. These are the same as analysed previously [6]. Figure 1A shows a few examples of nuclear images.

The median number of nuclear images was 281 (interquartile range (IQR), 273 to 289) for an ovarian carcinoma patient, and the median number of nuclear pixels in each image was about 3600 (IQR, 2700 to 4600). There was a median of 997 (IQR, 502 to 1452) nuclear images for each endometrial carcinoma patient, with a median size of about 4100 (IQR, 3200 to 5100) pixels. In total, 840,000 nuclear images were analysed.

### 2.3. Quantification of Diversity of Chromatin Compartments

#### 2.3.1. Segmentation of Chromatin Compartments

In order to segment chromatin compartments in the nuclear images, we generalised Niblack’s adaptive thresholding algorithm [21] and applied a metric inspired by Yanowitz and Bruckstein’s validation step [22] to automatically label each pixel as one of three categories based on the optical density of the pixel and the optical densities of nearby pixels. Appendix A describes relevant details, and Figure 1B illustrates some segmentation examples. The three categories were termed dark, grey, and bright. Regions labelled as dark or bright corresponded to relatively highly or relatively weakly condensed chromatin compartments, respectively, possibly heterochromatin and euchromatin. Pixels with intermediate optical densities were labelled grey and not analysed directly. The inclusion of this category facilitated separation of highly and weakly condensed compartments from the average optical density of nearby pixels, as opposed to separating highly and weakly condensed compartments from each other.

#### 2.3.2. Analysis of Chromatin Compartments

The size of a chromatin compartment was measured as the number of pixels in the segmented region, and the distribution of sizes in a nuclear image was found separately for dark regions and bright regions (Figure 1C). Similarly, the distribution of optical densities within dark regions and bright regions was found (Figure 1C). The entropy of each distribution was computed and then summed, resulting in a continuous value that described the diversity of either highly or weakly condensed chromatin compartments in the nucleus. Details of the computation are described in Appendix A, which is placed in the context described by Maître et al. [23] and Tupin et al. [24].

#### 2.3.3. Dual Entropy Sum Histogram

The entropy sum of each nuclear image was adaptively quantified separately for the dark and bright regions (see Appendix A for details). The resulting distributions of entropy sums for a patient are two sum histograms [25] (Figure 1D), which encapsulate information about the diversity of chromatin compartments in its cancer cell nuclei. We denoted each distribution as a dual entropy sum histogram (DESH) and represented each patient by the DESH of dark regions and the DESH of bright regions.

#### 2.3.4. Marker of Diversity of Chromatin Compartments

Using an established adaptive feature extraction method initiated by Walker et al. [26] and later developed by Albregtsen et al. [27,28,29,30], a single continuous value was computed from each DESH (see Appendix A for details). To simplify the two DESH features of a patient into an assessment of diverse vs. similar chromatin compartments, we tested a series of classification methods in the training subset of the ovarian carcinoma cohort, including parametric methods assuming Gaussian distributions, non-parametric methods such as Parzen window and *k* nearest neighbour, and support vector machines (SVMs), both linear SVM and with the Gaussian radial basis function. It appeared that the most prognostic classifier was produced by assuming equal prior probabilities and a Gaussian feature distribution in each outcome class with common covariance matrix. This simple and robust classification method [31,32] was therefore utilised to define a marker of diversity of chromatin compartments in both training subsets (Figure 1E). Figure 1 illustrates the major steps involved in assessing the novel marker for a new patient. Through the quantification of diversity of chromatin compartments in each patient’s tumour nuclei, the marker classifies the patient as having either diverse chromatin compartments (DCC) or similar chromatin compartments (SCC), indicating poor or good clinical outcome, respectively.

### 2.4. Statistical Analyses

Sensitivity and specificity on the test subset were calculated using the same definition of dichotomous clinical outcome as used for training. The classification accuracy was mainly measured in terms of the balanced correct classification rate (BCCR), i.e., the mean of sensitivity and specificity. This metric is also known as balanced accuracy and better describes a classifier’s ability to distinguish outcome groups of substantially different sizes than the ordinary correct classification rate (CCR). Survival analyses were performed using the same clinical outcomes as end points, but without the dichotomisation. The end points were defined as proposed by Punt et al. [33] and the same as in previous studies that analysed the same cohorts [9,19]. Specifically, time to recurrence was analysed for ovarian carcinoma patients and calculated from surgery to recurrence or 31st of December 1998. Cancer-specific survival was analysed for endometrial carcinoma patients and calculated from surgery to death from endometrial carcinoma or end of follow-up. Survival analyses were performed using the Cox proportional hazards model and Wald χ^2^ test, except for univariable survival analyses of categorical markers, which were performed using the log-rank test. The variables included in the multivariable models were the same as in previous studies that have analysed the same cohorts. Specifically, the multivariable analysis of ovarian carcinoma patients was adjusted for FIGO stage (IB or IC vs. IA) and histological grade (grade 3 or clear cell vs. grade 1 or 2) [6,9,16,34], while the multivariable analysis of endometrial carcinoma patients was adjusted for the preoperatively available markers age (continuous) and pathological risk classification (high-risk vs. low-risk) [6,19,34]. The pathological risk classification was based on the curettage histology report and assigned benign, experiencing hyperplasia or endometrioid grade 1 or 2 tumours as low-risk, while other tumours (i.e., endometrioid grade 3 or non-endometrioid) were assigned as high-risk [17]. Patients with missing data for variables included in a particular analysis were excluded from that analysis. A two-sided *p* < 0.05 was considered statistically significant. Texture and survival analyses were performed in MATLAB R2012b (The MathWorks, Natick, MA, USA) and Stata/SE 16.1 (StataCorp, College Station, TX, USA), respectively.

### 2.5. Data Availability Statement

Restrictions apply to the availability of the data presented in this study because patients did not provide permission for data sharing outside the institution or established collaborations.

## 3. Results

### 3.1. Patient Characteristics

Characteristics of the studied patients are presented in Appendix A. In the ovarian carcinoma cohort, 37% patients were classified as having DCC and 31% patients were classified as CHE. In the endometrial carcinoma cohort, 24% patients were classified as having DCC and 15% patients were classified as CHE. The median follow-up time was 11.7 (IQR, 8.3 to 13.6) years for the ovarian carcinoma patients, while it was 2.96 (IQR, 1.47 to 4.44) years for the endometrial carcinoma patients.

### 3.2. Size and Number of Chromatin Compartments

Patients with poor clinical outcome typically had more chromatin compartments per tumour cell nucleus than patients with good clinical outcome (Appendix A). Additionally, the segmented chromatin compartments were on average larger in patients with poor clinical outcome compared to patients with good clinical outcome (Appendix A). The same trends were observed for the relatively highly and the relatively weakly condensed compartments, i.e., the dark and the bright regions, respectively (Appendix A).

### 3.3. Relative Prognostic Value of Markers of Chromatin Entropy

The markers of chromatin entropy, e.g., chromatin heterogeneity and diversity of chromatin compartments, obtain their assessment from feature values that quantify prognostic indications on a continuous scale. The feature value of the marker of chromatin heterogeneity and the feature values of the marker of diversity of chromatin compartments were moderately strongly correlated in both test subsets (Figure 2). Similar correlations were also observed in the entire patient cohorts. In the ovarian carcinoma cohort, Pearson correlation coefficient was 0.68 (95% CI, 0.61 to 0.74) and 0.71 (95% CI, 0.64 to 0.76) when applying the feature value of the dark and bright regions, respectively. In the endometrial carcinoma cohort, Pearson correlation coefficient was 0.73 (95% CI, 0.70 to 0.76) and 0.73 (95% CI, 0.69 to 0.76) when applying the feature value of the dark and bright regions, respectively. In terms of classification, 417 (83%) of the 505 patients in the test subsets had tumours with either both SCC and CHO or both DCC and CHE. Only 21 (4%) had tumours with both SCC and CHE, while 67 (13%) had tumours with both DCC and CHO.

### 3.4. Classification Performance

The marker of diversity of chromatin compartments obtained a BCCR of 67% on both test subsets. This is four to five percentage points better than the marker of chromatin heterogeneity (Table 1). For reference, Table 1 also specifies performance estimates of the GLEM4D-based classifiers that were trained and tested by Nielsen et al. [9] and Hveem et al. [19] using the same train and test subsets as in the present study. Compared to these cancer type specific markers, the marker of diversity of chromatin compartments provided a two to three percentage points better BCCR on both test subsets (Table 1). Similar trends were observed for the train subsets (Table 1). All markers performed slightly better on the train subsets than the test subsets (Table 1), even though the marker of chromatin heterogeneity did not utilise the train subsets for training. Scatter plots of the two feature values of the marker of diversity of chromatin compartments illustrate good separation of the clinical outcome classes in both train and test subsets (Appendix A).

### 3.5. Survival Analyses of Entire Cohorts

Kaplan–Meier survival curves according to the marker of diversity of chromatin compartments are shown in Figure 3A,B. Estimated 10 year recurrence-free survival was 83% (95% confidence interval (CI), 76% to 88%) for ovarian carcinoma patients classified as having SCC and 49% (95% CI, 38% to 59%) for ovarian carcinoma patients classified as having DCC. For endometrial carcinoma patients, estimated 5 year cancer-specific survival was 91% (95% CI, 87% to 94%) for tumours with SCC and 67% (95% CI, 58% to 75%) for tumours with DCC.

The hazard ratio (HR) of recurring ovarian cancer was nearly four times higher for tumours with DCC compared to tumours with SCC (Table 2), and the HR of endometrial cancer-specific mortality was close to five times higher (Table 3). In both cohorts, the marker of diversity of chromatin compartments was significantly associated with clinical outcome after also adjusting for established clinical and pathological markers (Table 2 and Table 3). The adjusted HR of ovarian cancer recurrence and endometrial cancer-specific mortality was more than twice for tumours with DCC compared to tumours with SCC (Table 2 and Table 3). When analysing only the 767 endometrial carcinoma patients treated with hysterectomy, the HR was 5.2 (95% CI, 3.2 to 8.3) in univariable analysis and 3.1 (95% CI, 1.8 to 5.3) in multivariable analysis.

### 3.6. Survival Analyses in Subgroups of Chromatin Heterogeneity

In each patient cohort, the marker of diversity of chromatin compartments provided additional prognostic information in patients with CHO tumours (Figure 4 and Figure 5). The HR was 4.8 (95% CI, 2.5 to 9.5) and 2.6 (95% CI, 1.4 to 4.8) in the CHO subgroup of the ovarian and endometrial carcinoma cohort, respectively, and 5.2 (95% CI, 2.0 to 13.6) and 2.6 (95% CI, 1.1 to 6.0) in the corresponding test subsets. The marker of diversity of chromatin compartments was not significant in ovarian carcinoma patients with CHE tumours (Figure 4), but significant in endometrial carcinoma patients with CHE tumours (Figure 5).

When analysing a multivariable model consisting of the two markers of chromatin entropy, the HR of tumours with DCC compared to tumours with SCC was 3.1 (95% CI, 1.7 to 5.7) in the ovarian carcinoma cohort and 3.3 (95% CI, 1.9 to 5.6) in the endometrial carcinoma cohort. In these multivariable analyses, the marker of chromatin heterogeneity was not significant in the ovarian carcinoma cohort (HR = 1.36; 95% CI, 0.76 to 2.44; *p* = 0.29), but was significant in the endometrial carcinoma cohort (HR = 2.0; 95% CI, 1.2 to 3.5; *p* = 0.012). Analysing only endometrial carcinoma patients treated with hysterectomy slightly increased the prognostic difference between the two markers; HR was 3.8 (95% CI, 2.2 to 6.8) for tumours with DCC compared to tumours with SCC, while 1.8 (95% CI, 1.0 to 3.3) for CHE tumours compared to CHO tumours.

For reference, the corresponding comparisons were performed for the GLEM4D-based classifiers that were trained and tested by Nielsen et al. [9] and Hveem et al. [19] using the same train and test subsets as used for the marker of diversity of chromatin compartments. Similar trends and identical assessments of statistical significance were observed for the cancer type specific GLEM4D-based markers as for the pan-cancer marker of chromatin heterogeneity, both in subgroup analyses (Figure 4 and Figure 5) and in multivariable analyses with the marker of diversity of chromatin compartments.

### 3.7. Survival Analyses in Subgroups of Clinical and Pathological Markers

The relation to clinical and pathological risk factors could be better understood by analysing the marker of diversity of chromatin compartments in the subgroups defined by another patient characteristic. The forest plots in Figure 4 and Figure 5 indicate that the marker of diversity of chromatin compartments might have prognostic value in most of the analysed subgroups, albeit statistical significance was not consistently observed in subgroups with relatively few patients or events. Highest HRs were often observed in subgroups associated with low or medium risk, e.g., in terms of FIGO stage, histological grade or pathological risk classification.

When stratified on pathological risk classification, the HR of tumours with DCC compared to tumours with SCC was nearly two in analysis of the subgroup of patients classified as having high pathological risk (HPR) (Figure 4 and Figure 5). The prognostic value appeared higher in patients classified as having low pathological risk (LPR), where the HR associated with the marker of diversity of chromatin compartments was about five (Figure 4 and Figure 5). We integrated the two markers into a three-category risk stratification by defining low-risk tumours as those with both SCC and LPR, high-risk tumours as those with both DCC and HPR, and medium-risk tumours as those where the two markers indicated different prognoses. This three-level categorisation appeared to encapsulate the prominent survival differences identified by the two individual markers (Figure 3C,D). Distinctly different prognoses were observed in the three risk groups. The estimated 10 year recurrence-free survival of ovarian carcinoma patients was 95% (95% CI, 88% to 98%) in the low-risk group, 67% (95% CI, 55% to 76%) in the medium-risk group, and 42% (95%, 31% to 53%) in the high-risk group. Similarly, the estimated 5 year endometrial cancer-specific survival was 93% (95% CI, 90% to 96%) in low-risk, 76% (95% CI, 67% to 83%) in medium-risk, and 55% (95% CI, 40% to 68%) in high-risk. The survival estimates were slightly higher when excluding endometrial carcinoma patients not treated with a hysterectomy, becoming 94% (95% CI, 90% to 96%) in low-risk, 79% (95% CI, 70% to 86%) in medium-risk, and 58% (95% CI, 42% to 71%) in high-risk.

Stage I endometrial carcinoma patients may post-surgically be risk stratified based on histological type and grade, substage and lymphovascular space invasion status [36]. In stage I patients classified as high-risk, the 51 patients with DCC had an estimated 5 year endometrial cancer-specific survival of 69% (95% CI, 51% to 82%), compared to 98% (95% CI, 86% to 100%) for the 47 patients in this subgroup with SCC. The corresponding HR was 11.4 (95% CI, 1.5 to 87.6). Only 10 (2%) cancer-specific deaths were observed for the 508 stage I patients with low, intermediate, or high-intermediate risk. For these patients, the estimated 5 year endometrial cancer-specific survival was 95% (95% CI, 81% to 99%) for the 65 patients with DCC and 97% (95% CI, 94% to 99%) for the 443 patients with SCC. The corresponding HR was 2.24 (95% CI, 0.46 to 10.80).

## 4. Discussion

We applied texture analysis methods to analyse the chromatin compartments in cancer cell nuclei and automatically quantify the diversity of chromatin compartments in each nucleus. In patients with gynaecological carcinomas, poor clinical outcome was observed to be associated with both more and larger chromatin compartments than good clinical outcome. This finding in established cancer extends previous reports on the changes in size and number of chromatin compartments that occur during carcinogenesis [10,11,12,13,14,15]. We observed the increase in number and size for both the relatively highly and the relatively weakly condensed compartments, suggesting that poor clinical outcome is associated with cancer cell nuclei where the chromatin is more segregated into heterochromatic and euchromatic regions, as opposed to being a mixture of the two. The entropy of the compartment sizes and the entropy of the optical densities within compartments were utilised by robust machine learning algorithms to obtain a marker of diversity of chromatin compartments. The marker accurately predicted ovarian cancer recurrence and endometrial cancer-specific mortality, providing HRs of about four and a mean of sensitivity and specificity of 67% on test subsets, which further suggests that diversity of chromatin compartments is a characteristic of poor prognosis.

The pan-cancer marker of chromatin heterogeneity applied a purely statistical texture analysis method to identify chromatin patterns associated with poor clinical outcome [6] and could potentially be applicable to guide the selection of adjuvant treatments in a range of cancer types [37]. The combined structural and statistical texture analysis method applied by the marker of diversity of chromatin compartments provides a different approach to quantify chromatin entropy. The prognostic value of the novel marker correlated moderately strongly with the prognostic value of the marker of chromatin heterogeneity, but the novel marker was also observed to provide additional prognostic information. Moreover, the marker of diversity of chromatin compartments appeared to provide prognostic information supplementary to established prognostic factors. Subgroup analyses suggested that the added prognostic information was largest in patient groups associated with low or medium risk in terms of pathological markers.

In the clinic, multiple prognostic factors are often applied to inform the discussion with patients on the risks and benefits of different options for adjuvant treatment. In ovarian carcinoma, chemotherapy is generally recommended in stage II to IV, while no adjuvant chemotherapy, platinum-based monotherapy or combination chemotherapy may all be appropriate options in stage I [38]. The analysed pathological risk classification for stage I ovarian carcinoma combines FIGO substage and histological grade [35], and the resulting high-risk patients have been shown to benefit from adjuvant chemotherapy [39]. In our patient cohort, 10 year recurrence-free survival of ovarian carcinoma was 53% if labelled high-risk by the pathological risk classification and otherwise 91%. This survival difference is comparable with that of the marker of diversity of chromatin compartments, and the two markers each provided supplementary prognostic information that allowed integration into a three-category risk assessment. The finer risk stratification observed with this novel categorisation suggests that it could possibly be used to inform the discussion of adjuvant treatment with patients with stage I ovarian carcinoma. Further investigations are warranted to assess the effect of adjuvant treatment on these risk groups.

The high accuracy of the marker of diversity of chromatin compartments on the endometrial carcinoma cohort shows that the marker can be reliably assessed using curettage specimens, not only when using resection specimens, which were used for the ovarian carcinoma cohort. The possibility to preoperatively risk-stratify patients could reduce morbidity from extensive surgery of low-risk endometrial cancer patients [40] and could also indicate the viability of fertility-sparing surgery. The novel marker provided prognostic information supplementary to the pathological risk classification based on the curettage histology. Integration of the two markers provided a three-category risk assessment, which was observed to identify endometrial carcinoma patients with distinctly different probabilities of cancer-specific survival. The consequence of using this risk stratification to guide the choice of surgical procedure may be assessed in a randomised controlled trial.

About 80% of endometrial cancer patients are diagnosed with FIGO stage I tumours [36]. The recommendations of adjuvant treatment for these patients range from no adjuvant treatment and only vaginal cuff brachytherapy to complete pelvic radiation and possibly chemotherapy [36,41]. In particular, vaginal cuff brachytherapy or pelvic radiation has been recommended for patients classified as high-risk (i.e., endometrioid stage IB grade 3 or non-endometrioid stage I), and adjuvant chemotherapy may also be considered or recommended if no surgical nodal staging was performed or the tumour is non-endometrioid [36,41]. The marker of diversity of chromatin compartments appeared strongly prognostic in this patient subgroup. The excellent cancer-specific survival rate observed for the patients in this subgroup with SCC suggests that adjuvant treatment might not be needed for these patients, which would spare patients from the associated morbidities, while the low survival rate observed for the corresponding patients with DCC suggests that it might be appropriate to offer these patients adjuvant chemotherapy. This interesting finding warrants further investigations, in particular in a randomised controlled trial to assess its clinical utility.

## 5. Conclusions

We developed a novel marker of diversity of chromatin compartments by segmenting and analysing the chromatin compartments in nuclei from gynaecological carcinomas. Patients with poor clinical outcome were associated with both more and larger chromatin compartments than patients with good clinical outcome. The marker correlated with the marker of chromatin heterogeneity but was observed to provide prognostic information supplementing chromatin heterogeneity and other prognostic markers. Integration with pathological risk classification provided three patient groups associated with substantially different risks of ovarian cancer recurrence and endometrial cancer-specific mortality. Clinical application of the novel risk assessments might be used to tailor surgical and adjuvant treatments to individual gynaecological carcinoma patients. This could possibly reduce overtreatment of low-risk patients and facilitate more comprehensive adjuvant treatments to be offered to high-risk patients. However, randomised controlled trials must first be conducted to assess the efficacy of treatments in the different risk groups.

## Figures and Tables

**Figure 1 cancers-12-03838-f001:**
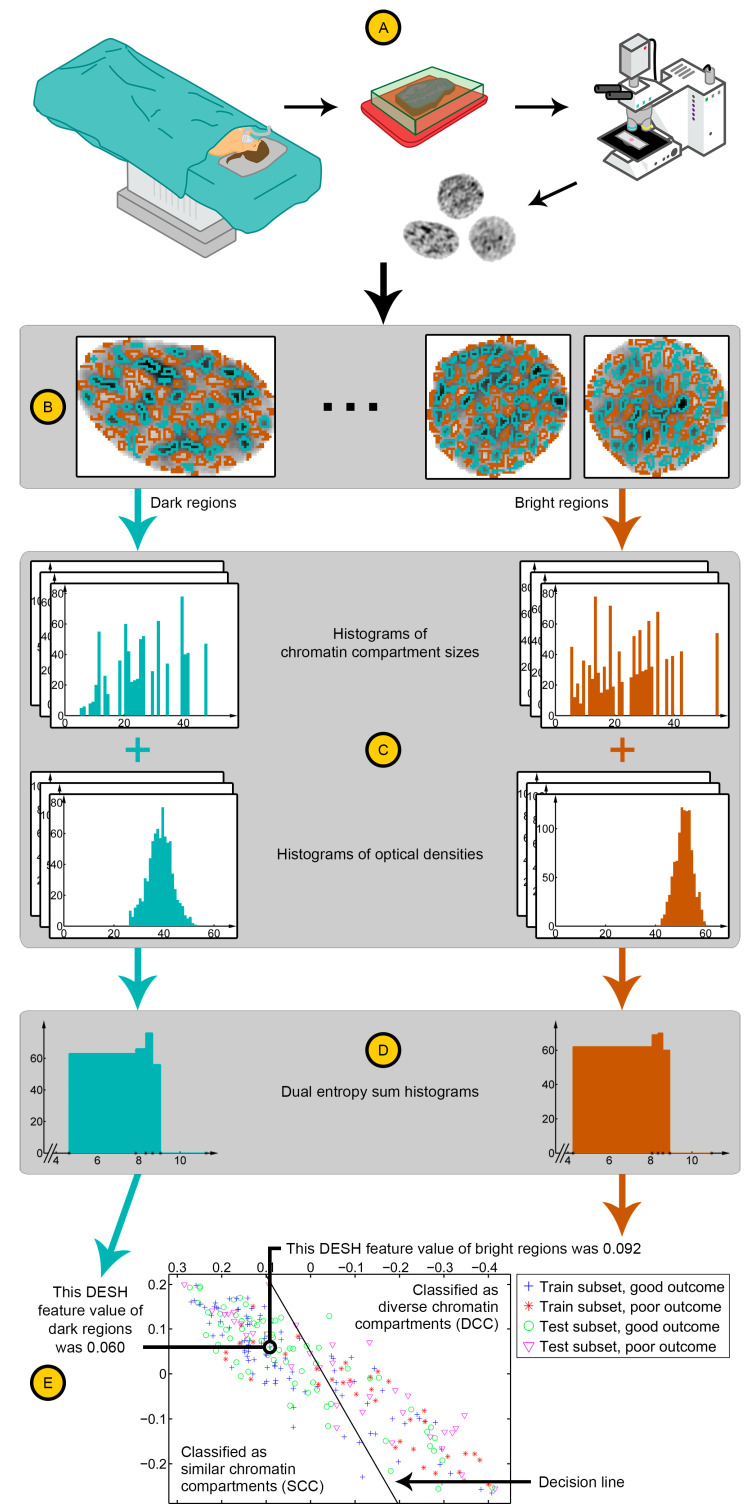
Pipeline for assessing the marker of diversity of chromatin compartments. (**A**) Tumour cell nuclei from a formalin-fixed and paraffin-embedded (FFPE) block are imaged by a microscope. (**B**) The segmentation algorithm automatically finds dark (shown in cyan) and bright (shown in light brown) regions, corresponding to relatively highly and relatively weakly condensed chromatin compartments, respectively. (**C**) The histogram of chromatin compartment sizes and the histogram of optical densities in the compartments are computed separately for the dark and bright regions in the nuclear image. The entropy of each histogram is then summed, giving a single continuous value that describes the diversity of dark chromatin compartments in the nuclear image, and similarly a value that describes the bright chromatin compartments. (**D**) The entropy sums of a patient’s nuclear images provide the dual entropy sum histogram (DESH) of the dark regions and the DESH of the bright regions. (**E**) The two adaptive DESH features are computed, one from each DESH. The classification of a patient, as having either diverse chromatin compartments (DCC) or similar chromatin compartments (SCC), is determined by comparing the two features to the decision line. DCC, diverse chromatin compartments; DESH, dual entropy sum histogram; FFPE, formalin-fixed and paraffin-embedded; SCC, similar chromatin compartments.

**Figure 2 cancers-12-03838-f002:**
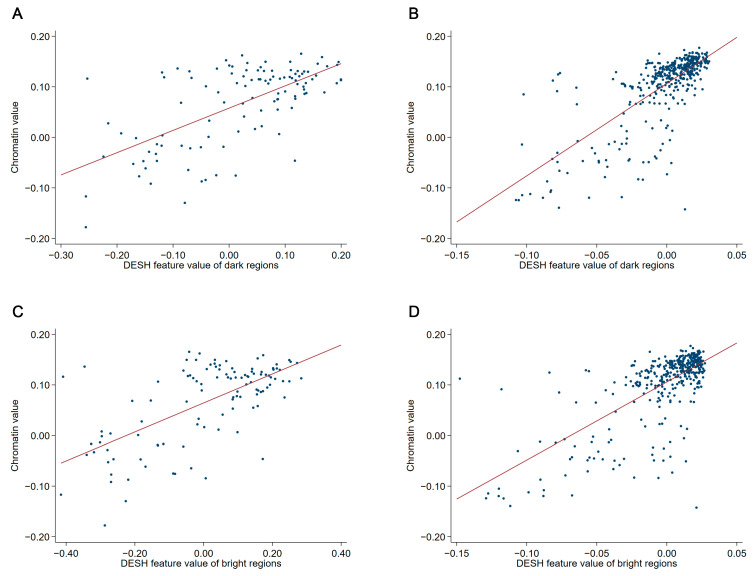
Scatter plots of the chromatin value, i.e., feature value of the marker of chromatin heterogeneity, and the feature values of the marker of diversity of chromatin compartments. (**A**) The chromatin value vs. the DESH feature value of dark regions in the test subset of the ovarian carcinoma cohort. Pearson correlation coefficient was 0.64 (95% CI, 0.51 to 0.74). (**B**) The chromatin value vs. the DESH feature value of dark regions in the test subset of the endometrial carcinoma cohort. Pearson correlation coefficient was 0.73 (95% CI, 0.68 to 0.77). (**C**) The chromatin value vs. the DESH feature value of bright regions in the test subset of the ovarian carcinoma cohort. Pearson correlation coefficient was 0.65 (95% CI, 0.52 to 0.74). (**D**) The chromatin value vs. the DESH feature value of bright regions in the test subset of the endometrial carcinoma cohort. Pearson correlation coefficient was 0.70 (95% CI, 0.64 to 0.75). DESH, dual entropy sum histogram.

**Figure 3 cancers-12-03838-f003:**
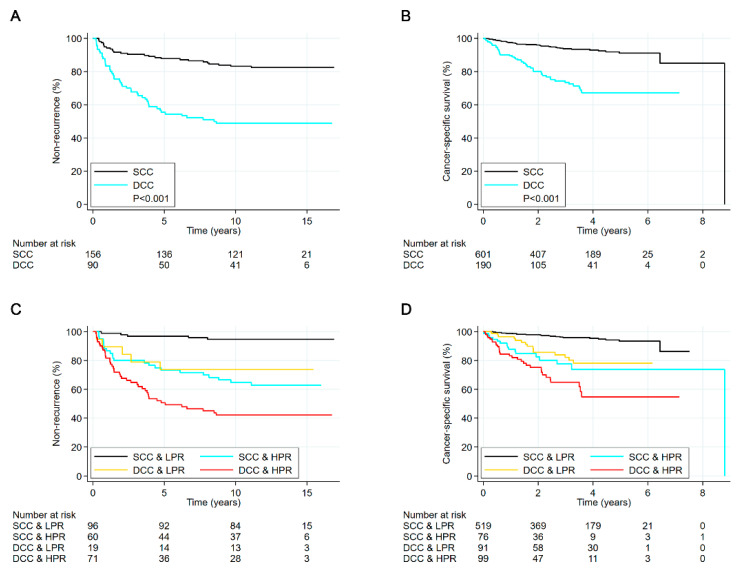
(**A**) Kaplan–Meier curves by marker of diversity of chromatin compartments in the ovarian carcinoma cohort. (**B**) Kaplan–Meier curves by marker of diversity of chromatin compartments in the endometrial carcinoma cohort. (**C**) Kaplan–Meier curves by marker of diversity of chromatin compartments and pathological risk classification in the ovarian carcinoma cohort. Pathological high-risk tumours were defined as those with either clear cell histology, poor differentiation or both moderate differentiation and International Federation of Gynecology and Obstetrics (FIGO) stage IB or IC, while other tumours were defined as low or medium risk [35]. (**D**) Kaplan–Meier curves by marker of diversity of chromatin compartments and pathological risk classification in the endometrial carcinoma cohort. Pathological high-risk tumours were defined as those assessed as non-endometrioid or endometrioid grade 3 using the curettage specimens, while low-risk if benign, experiencing hyperplasia or endometrioid grade 1 or 2 [17]. DCC, diverse chromatin compartments; HPR, pathological risk classification was high; LPR, pathological risk classification was low (or medium); SCC, similar chromatin compartments.

**Figure 4 cancers-12-03838-f004:**
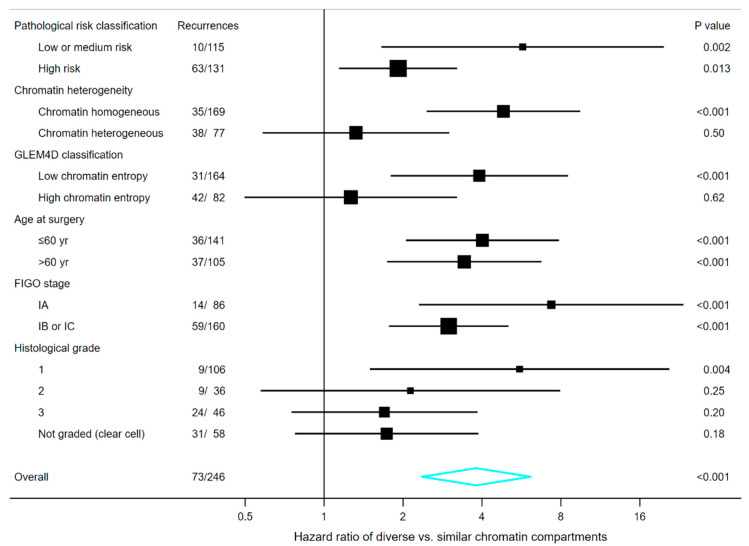
Forest plot illustrating the hazard ratio and 95% confidence interval of the marker of diversity of chromatin compartments in subgroups of patients with ovarian carcinoma. Pathological high-risk tumours were defined as those with either clear cell histology, poor differentiation or both moderate differentiation and FIGO stage IB or IC, while other tumours were defined as low or medium risk [35]. GLEM4D classifications refer to the marker trained and tested by Nielsen et al. [9] using the same ovarian carcinoma cohort as the marker of diversity of chromatin compartments. FIGO, International Federation of Gynecology and Obstetrics; GLEM4D, four-dimensional grey level entropy matrix.

**Figure 5 cancers-12-03838-f005:**
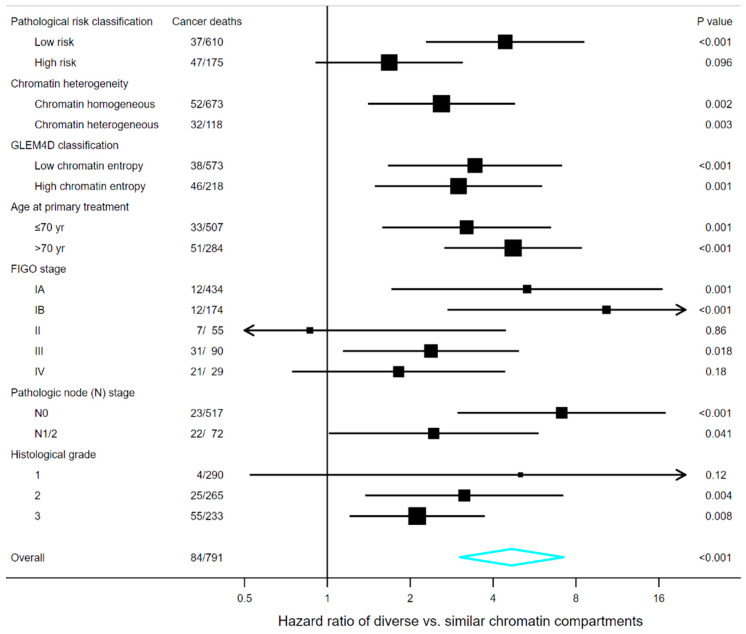
Forest plot illustrating the hazard ratio and 95% confidence interval of the marker of diversity of chromatin compartments in subgroups of patients with endometrial carcinoma. Pathological high-risk tumours were defined as those assessed as non-endometrioid or endometrioid grade 3 using the curettage specimens, while low-risk if benign, experiencing hyperplasia or endometrioid grade 1 or 2 [17]. The hazard ratio is not depicted for the subgroup of patients with chromatin heterogeneous tumours because no cancer-specific death occurred among the 26 patients classified as having similar chromatin compartments, and thus the estimated hazard ratio approached infinity (there were 32 events among the 92 patients classified as having diverse chromatin compartments). *GLEM4D classifications* refer to the marker trained and tested by Hveem et al. [19] using the same endometrial carcinoma cohort as the marker of diversity of chromatin compartments. FIGO, International Federation of Gynecology and Obstetrics; GLEM4D, four-dimensional grey level entropy matrix.

**Table 1 cancers-12-03838-t001:** Classification performance of markers of chromatin entropy.

		Ovarian Carcinoma				Endometrial Carcinoma			
Marker	Set	BCCR	CCR	Sens.	Spec.	BCCR	CCR	Sens.	Spec.
Diversity of chromatin compartments									
	Train	71.7%	72.4%	70.3%	73.2%	67.5%	77.6%	54.5%	80.5%
	Test	66.8%	69.9%	57.1%	76.5%	66.9%	76.3%	55.0%	78.8%
	All	69.2%	71.3%	63.9%	74.5%	67.2%	77.0%	54.8%	79.6%
Chromatin heterogeneity									
	Train	65.7%	70.9%	54.1%	77.3%	64.0%	81.9%	40.9%	87.0%
	Test	63.2%	67.0%	51.4%	75.0%	61.8%	83.2%	35.0%	88.7%
	All	64.6%	69.2%	52.8%	76.4%	63.0%	82.6%	38.1%	87.8%
GLEM4D (for specific cancer type)									
	Train	70.1%	72.4%	64.9%	75.3%	66.0%	74.9%	54.5%	77.4%
	Test	63.9%	68.0%	51.4%	76.5%	64.5%	72.0%	55.0%	73.9%
	All	67.0%	70.5%	58.3%	75.8%	65.2%	73.5%	54.8%	75.7%

BCCR, balanced correct classification rate; CCR, correct classification rate; GLEM4D, four-dimensional grey level entropy matrix.

**Table 2 cancers-12-03838-t002:** Analyses of time to recurrence for patients with ovarian carcinoma.

		Univariable Analysis		Multivariable Analysis	
Marker	Variable Treatment	HR (95% CI)	*p* Value	HR (95% CI)	*p* Value
Diversity of chromatin compartments	DCC vs. SCC	3.8 (2.4–6.1)	<0.001	2.1 (1.3–3.5)	0.004
FIGO stage	IB or IC vs. IA	2.6 (1.5–4.7)	0.001	2.1 (1.2–3.8)	0.014
Histological grade	Grade 3 or clear cell vs. Grade 1 or 2	5.6 (3.3–9.6)	<0.001	4.0 (2.2–7.1)	<0.001

CI, confidence interval; DCC, diverse chromatin compartments; FIGO, International Federation of Gynecology and Obstetrics; HR, hazard ratio; SCC, similar chromatin compartments.

**Table 3 cancers-12-03838-t003:** Analyses of cancer-specific survival for patients with endometrial carcinoma.

		Univariable Analysis		Multivariable Analysis	
Marker	Variable Treatment	HR (95% CI)	*p* Value	HR (95% CI)	*p* Value
Diversity of chromatin compartments	DCC vs. SCC	4.7 (3.0–7.2)	<0.001	2.4 (1.5–3.9)	0.001
Pathological risk classification	High risk vs. Low risk	5.8 (3.8–9.0)	<0.001	3.3 (2.0–5.4)	<0.001
Age at primary treatment	1-year increment	1.06 (1.04–1.09)	<0.001	1.05 (1.03–1.07)	<0.001

CI, confidence interval; DCC, diverse chromatin compartments; HR, hazard ratio; SCC, similar chromatin compartments.

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
