# Peer review of "Prognostic Value of the Diversity of Nuclear Chromatin Compartments in Gynaecological Carcinomas"

_cancers, 2020, doi:10.3390/cancers12123838_

Round 1

Reviewer 1 Report

Kleppe et al. performed a texture analysis method and machine learning algorithms to quantify the diversity of chromatin compartments in patient with gynaecological carcinomas. They observe that increased in size and number of dense and light chromatin significantly associates with poor prognosis gynaecological carcinomas.

Image analysis pipeline is well designed and statistical analysis accurate.

The new marker they define, which however needs to be assessed in randomize controlled trials, allows to better stratified patient and may help refine personalized treatments

I recommend publication.

Author Response

Many thanks for the positive comments.

Reviewer 2 Report

In this study, the authors wanted to explore in depth the information derived from the analysis of the state and organization of nuclear chromatin in ovarian and endometrial cancers. The scientific/clinical interest of the general theme of this work, i.e. the advanced analysis of images obtained from FFPE blocks in order to obtain new high-performance prognostic and predictive markers, is obvious. This is a well-written manuscript, all necessary is reported. The size of the cohorts (246 patients for ovarian cancer and 791 patients for endometrial cancer), as well as the very high number of images analyzed per patient (281 for ovarian cancer and 997 for endometrial cancer) should be highlighted, which gives weight to the work presented. The data presented are very thorough with appropriate statistical analysis. The results are very interesting with this new marker of the diversity of chromatin compartments. It is relevant to note that, incorporating the pathological risk classification has enabled the authors to obtain three groups of patients with significantly differential risks of ovarian cancer recurrence and mortality specific to endometrial cancer. The new data obtained here, after confirmation in randomized clinical trials, will make it possible to see if this marker could improve the clinical management of the 3 identified groups.

Author Response

Many thanks for the positive comments. We have amended lines 35-38 in the revised manuscript to highlight the median number of nuclei analysed for each patient.

Reviewer 3 Report

In this study the authors propose to use measurements of entropy in nuclear subregions to examine whether they can assist in prognosis of cancer cells. Previous studies have used entropy to measure the heterogeneity of chromatin regions. Here, they want to also examine the diversity of chromatin compartments in the nucleus of cancer cells, and in comparison to regions of different chromatin densities. The cells used in the study are from patients with ovarian carcinoma patients and endometrial carcinoma. They find that cells from patients with poor clinical outcome usually had more chromatin compartments per nucleus than those with good clinical outcome and that the segmented chromatin regions were larger in cells from patients with poor clinical outcome and this correlated to the 'dark' and 'bright' regions they define. In addition, they suggest that poor clinical outcome for the patients is associated with nuclei in where the chromatin seems segregated into heterochromatic and euchromatic regions rather than being a mixture of subcompartments. This might be a prognostic marker. The paper is well written and explained and deserves publication.

Author Response

Many thanks for the positive comments.